# On the Possible Cause of Sudden Storage Modulus Increase during the Heating of PM FeMnSiCrNi SMAs

**DOI:** 10.3390/nano12142342

**Published:** 2022-07-08

**Authors:** Bogdan Pricop, Marian Grigoraș, Firuța Borza, Burak Özkal, Leandru-Gheorghe Bujoreanu

**Affiliations:** 1Faculty of Materials Science and Engineering, “Gheorghe Asachi” Technical University of Iași, Blvd. Dimitrie Mangeron 71A, 700050 Iași, Romania; bogdan.pricop@academic.tuiasi.ro; 2National Institute of Research and Development for Technical Physics, Blvd. Dimitrie Mangeron 47, 700050 Iași, Romania; mgrigoras@phys-iasi.ro (M.G.); fborza@phys-iasi.ro (F.B.); 3Particulate Materials Laboratory, Metallurgical and Materials Engineering Department, Istanbul Technical University, 34469 Istanbul, Turkey; ozkal@itu.edu.tr

**Keywords:** FeMnSi-based shape memory alloys, storage modulus, internal friction, magnetization, *α’* martensite, martensite plates, lattice parameter, pre-strain, austenite

## Abstract

A sudden increase in storage modulus (*ΔE′*) was repeatedly recorded during the heating of powder metallurgy (PM) 66Fe-14Mn-6Si-9Cr-5Ni (mass. %) shape memory alloy specimens subjected to dynamic mechanical analysis (DMA), under constant applied strain amplitude and frequency. This instability, exceeding 12 GPa, was associated with the reverse martensitic transformation of *α′*-body centered cubic (bcc) martensite to *γ*-face centered cubic (*fcc*) austenite, overlapped on a magnetic transition. This transition, observed by thermomagnetic measurements (T-MAG), was associated with the temporary spontaneous alignment of magnetic spins, which lasted until thermal movement became prevalent, during heating. *ΔE′* was located around 250 °C on DMA thermograms and this temperature had the tendency to increase with the solution treatment temperature. On T-MAG diagrams, magnetization saturation temperature decreased from 405 °C to 52 °C with the increase in applied magnetic field from 20 Oe to 1 kOe and the increase in mechanically alloyed powder volume from 20% to 40%. On scanning electron micrographs, the presence of thermally induced *α’*-bcc martensite was emphasized together with the sub-bands that impede its stress-induced formation during DMA solicitation. On X-ray diffraction patterns of the solution-treated specimens, the presence of 22–82% *α′*-bcc martensite was identified, together with 8–55% retained austenite. It was assumed that the pre-existence of austenite together with *α’*-bcc martensite, in the microstructure of the solution-treated specimens, favored the magnetic transition, which destabilized the material and caused the storage modulus increase. The specimen comprising the largest amounts of austenite experienced the largest *ΔE′*.

## 1. Introduction

FeMnSiCrNi shape memory alloys (SMAs) represent a distinctive group within FeMnSi-based SMA systems, relaying on the thermally induced reversion to *γ*-face centered cubic (*fcc*) paramagnetic austenite of stress induced ε-hexagonal close packed (*hcp*) paramagnetic martensite, which occurs between the critical temperatures *A_s_* and *A_f_* [1]. These alloys typically comprise low manganese contents (around 14 mass. %), which enhance the formation of *α’*-body centered cubic (bcc) ferromagnetic martensite, especially after the application of high deformation degrees [2]. The morphology of *ε*–hcp martensite plates is typically linear and the plates have a triangular disposal, according to the close-packed habit planes of the *{111}_γ_* type [3]. On the other hand, *α’*-bcc martensite typically forms at the intersection of *ε*–hcp martensite plates [4] and, for this reason, they are shorter, have a lath or lenticular shape and do not entirely cross austenite grains [5]. These features are associated with an obstruction of reverse martensitic transformation, thus causing a diminution of shape memory effect (SME) [6]. In FeMnSi-based SMAs, it was argued that martensite reversion to austenite, during heating, must occur prior to the antiferromagnetic–paramagnetic transition produced at Néel temperature (*T_N_*) [7]. In other words, the antiferromagnetic–paramagnetic transition must occur at lower temperatures than the martensite transformation, because the two phenomena are competitive and austenite becomes stabilized by antiferromagnetic ordering and no longer transforms to martensite on cooling [8]. Several experimental techniques have been used to determine *T_N_*. Thus, during the heating-cooling cycles, *T_N_* corresponds to a maximum of magnetization [9] and an inflexion in the linear variations of electrical resistivity with temperature [10]. Peculiar variations of storage modulus (*E′*) and internal friction (*tan δ*) were observed by dynamic mechanical analysis (DMA) during the heating of FeMnSi-based SMAs. Storage modulus normally experiences a sudden drop during reverse martensitic transformation [11], but there is also a small increase, located at lower temperatures and accompanied by slope change, which was ascribed to the antiferromagnetic–paramagnetic transition [12]. In addition to the major internal friction maximum, observed during heating, which has been commonly ascribed to the *ε → γ* reversion [13], another small intensity *tan δ* maximum was reported, which was associated with the paramagnetic ↔ antiferromagnetic transformation [14]. Its thermal range, located to lower temperatures than that of martensite reversion [15], was argued by the fact that the relaxation of magnetic ordering was produced at a higher rate than the atomic shuffles that caused martensitic transformation [6].

In the last decade, powder metallurgy (PM) associated with mechanical alloying (MA) was identified as a processing routine of FeMnSiCrNi SMAs, which was developed as an alternative to classical ingot metallurgy [16]. In PM-MA FeMnSiCrNi SMAs, the thermally induced formation of *α’*-bcc martensite was observed by XRD and SEM [17]. A local storage modulus increase (*ΔE′*) was revealed during DMA tests and its intensity experienced a decreasing tendency with the increase in solution treatment temperature [18]. High resolution SEM investigations revealed the occurrence of a prominent surface relief accompanying the formation of lower compactness bcc lattice of *α′* martensite within the *γ*—fcc austenite matrix [19]. Due to the high distortions of the austenite matrix, internal sub-bands may form within *α′* martensite that hinder the growth of stressed-induced martensite plates [20]. More recently, some of the present authors reported a low-intensity *tan δ* maximum between 190 and 299 °C and a high-intensity one between 259–377 °C, which were ascribed to the antiferromagnetic–paramagnetic transition and to reverse the martensitic transformation, respectively, and their corresponding temperatures experienced a decrease with increasing MA fraction [21].

The present paper aims to investigate further the sudden storage modulus increase occurring during the heating of PM-MA FeMnSiCrNi SMAs, intending to clarify the effect of antiferromagnetic–paramagnetic transitions on the reverse martensitic transformations of both *ε*—hcp and *α′*-bcc martensites.

## 2. Materials and Methods

In order to reach nominal chemical composition of 66Fe-14Mn-6Si-9Cr-5Ni (mass%), which was selected for this study, two different powder metallurgical routes were followed. In the first route, individual elemental powders (Fe, Mn, Si, Cr and Ni) were directly weighed and blended in a polymeric jar using a turbula blender (Glen Mills^TM^, T2C) (Glen Mills, Clifton, NJ, USA)for 30 min to obtain a homogeneous powder mixture under solid-state mixing conditions (M1). For the second route, the powder mixture (M1) was mechanically alloyed using high energy ball mill (SPEX^TM^ D8000) (SPEX SamplePrep, Metuchen, NJ, USA) under an argon atmosphere with stainless steel vials and stainless steel milling balls for a 4 h duration, to obtain the powder mixture (M2) under the previously detailed heavily deformed solid-state mixing conditions [22,23]. By using M1 and M2, the following three powder batches were prepared. As first batch, M1 powder mixture was directly used (hereafter designated as 0_MA) (Batch 1). In order to obtain the second and third batches, M1 and M2 were remixed in different amounts using a turbula blender under the above-mentioned conditions. While the second batch had 20 vol. % M2 addition (hereafter 20_MA) (Batch 2), the third batch had 40 vol. % M2 addition (hereafter 40_MA) (Batch 3).

The three batches were compacted at 500 MPa in rectangular prism molds, with a hydraulic press and sintered (1150 °C/2 h under Ar) with oxide reduction with hydrogen (800 °C/30 min) [24]. The specimens’ density was further increased by hot rolling performed at 1100 °C with a final pass at room temperature (RT). Thus, the specimens’ thickness decreased to approx. 1 mm and porosity degree decreased from 16.85% to 2.51% [25]. In order to relieve rolling stresses and to improve martensite formation, the samples were subjected to a solution treatment (ST) (700 and 1100 °C)/5 min/water [26]. Each specimen was further designated with its MA fraction and ST time (e.g., *20_MA_700* represents the specimen with 20 vol. % MA powders, solution treated at 700 °C). Rectangular specimens (1 mm × 4 mm × 25 mm) were cut by wire spark erosion and the superficial oxidized and demanganized layers of the specimens were ground under water cooling.

As previously pointed out, systematic measurements of chemical composition of PM-MA 66Fe-14Mn-6Si-9Cr-5Ni SMA revealed standard deviations ranging between 0.12 at% for Mn and 0.39 at % for Ni [27]. Due to the intense milling stresses, the formation of large amounts of thermally induced *α′*-bcc martensite has been favored [16]. Since *α′*-bcc martensite is the first to revert to *γ*-*fcc* austenite, during heating, the reversion of *ε*-hcp occurs at higher temperatures, which broadens the interval of critical transformation temperatures. Nevertheless, the alloys experienced free-recovery SME. For instance, a specimen with 46.8% *α′*-bcc and 25.5% *ε*-hcp developed a 24 mm stroke when heated to 208 °C [19]. By cinematographic analysis, the *A_s_* temperature of a trained *20_MA* specimen was estimated at 50 °C [21].

Mechanical, magnetic and structural characterization were performed by dynamic mechanical analysis (DMA), thermomagnetic measurements (T-MAG), scanning electron microscopy (SEM) and X-ray diffraction (XRD).

DMA investigations were performed on a NETZSCH DMA 242 Artemis analyzer (Netzsch, Selb, Germany), equipped with three-point-bending specimen holder, during the heating from room temperature (RT) to 400 °C, with 5 °C/min, a pushrod amplitude of 20 µm (0.02 strain amplitude) and bending frequency of 1 Hz. DMA thermograms display the variations of storage modulus (*E′*) and internal friction (*tan δ,* defined as the ratio between loss, *E″*, and storage modulus) with temperature [21].

T-MAG measurements were performed using a Vibrating Sample Magnetometer (VSM) (Lake Shore VSM 7410) (Lake Shore Cryotronics Inc., Westerville, OH, USA).

In a maximum applied field of 20 kOe. The hysteresis loops were measured at temperatures ranging from RT to 800 °C. The magnetization curves versus temperature, in the interval from 25 °C to 450 °C, were measured for different values of the applied magnetic fields from 20 Oe to 1 kOe. T-MAG diagrams present the variation of magnetization moment (*M*) as a function of temperature [19].

SEM micrographs were recorded both on a common device, VEGA II LSH TESCAN (TESCAN, Brno – Kohoutovice, Czech Republic) and on a high-resolution device (HR-SEM) Neon 40 Esb Scanning Electron Microscope (Carl Zeiss NTS, Oberkochen, Germany), resolution of 1.1 ÷ 2.5 nm at a voltage U = 20 ± 1 kV. The specimens were prepared by water-stream grinding and polishing, followed by etching with a solution of 1.2% K_2_S_2_O_5_ + 1%NH_4_HF_2_ in 100 mL distilled water [26]. Particular care was taken in order to neutralize etched surfaces, considering the degradability tendency of FeMnSi-based SMAs after the deterioration of protective outer oxide layer [28].

XRD patterns were registered on a diffraction interval 2θ = 40–100°, using a Bruker ASX D8 Advance (Bruker, Billerica, MA, USA) diffractometer with Cu K_α_ radiation and 00-034-0396, 01-071-8288 and 01-071-8285 JCPDS files to identify the metallographic phases *α′*-bcc, *γ*—*fcc* and *ε*—hcp [20].

## 3. Results and Discussion

### 3.1. Dynamic Mechanical and Thermomagnetic Analysis

Figure 1 overlaps the variations with temperature of *E′* and *tan δ* from DMA thermograms and *M*, under the effect of an applied magnetic field of 20 Oe, from T-MAG diagrams. All of them were recorded during heating.

Five of the six thermograms present two *tan δ* maxima, differentiated by peak intensity and temperature. Considering the above discussion and assuming that the specimens are in an initial martensitic state, it can be assumed that the former peak could correspond to *α′*-bcc reversion to *γ*–*fcc* austenite eventually overlapped with a magnetic transition, and the latter to *ε*–hcp martensite reversion to austenite.

In each case, *tan δ* values after the second peak, corresponding to fully austenitic state, were higher than those corresponding to martensitic state, since austenite has more closed-packed planes [29]. This phenomenon was ascribed to the modulus softening associated with the ε→γ transformation during heating [14], as it will be further detailed. Another feature, noticeable on each of the six thermograms, is the sudden increase in storage modulus (*ΔE′)*, exceeding 12 GPa, at specimen *20_MA_1100*, while the lowest value was 2.7 GPa for *40_MA_1100*. Since the martensitic transformation of FeMnSiCrNi SMA is not thermoelastic, the storage modulus of martensite is higher than that of austenite and, excepting for the above-mentioned sudden increase, *E′* continuously falls during heating. When considering the formula of internal friction, *tan δ* = *E″*/*E′*, it is expectable that the decrease in the storage modulus would cause a higher internal friction value. On the other hand, it was argued that, for PM-MA FeMnSiCrNi SMAs, a temperature gap between the variations of *E′* and *tan δ* currently occurs during heating due to porosity presence, which retards the elastic response of the material with respect to viscous response [21].

The variation of the third parameter displayed in Figure 1, magnetic moment (*M*), typically experienced a continuously increase during heating, which, up to a certain point, could be explicable by the intensification of atomic spin movement owing to a temperature increase, which enables magnetic moments to follow the direction of the magnetic field.

*T_N_* is currently identified on T-MAG curves by a magnetization maximum. Such a maximum is noticeable only in Figure 1b,d, being located at much higher temperatures as compared to the sudden storage modulus increase. Considering the technical recommendations of the experimental devices, the two experiments were performed at different heating rates: 2 °C/min for T-MAG and 5 °C/min for DMA. This heating rate difference, together with the low values of applied magnetic field, could be the cause for the temperature gap between the *E′* increase and *M* maximum.

On the other hand, the magnetization maxima, observed during heating, that have been associated with *T_N_*, are typically characterized by: (i) low magnetic moments, (ii) low temperatures and (iii) higher applied magnetic fields. Accordingly, the magnetization of the antiferromagnetic phase is one order of magnitude lower than that of the ferromagnetic phase [30], most of *T_N_* values are below 100 °C [31] and the applied magnetic field may increase up to 7 T [9]. These features can be associated with the presence of ferromagnetic *α′*-bcc martensite, as it will be shown later.

For a better insight on the effects of applied magnetic field, Figure 2 presents the magnetic hysteresis loops for the samples *20_MA_700* and *40_MA_700*, at different measurement temperatures. These two specimens were selected because they contain MA’ed powder and were solution-treated at the same temperature and one of them (*40_MA_700*) reached the highest value of magnetization moment (M).

The saturation magnetization decreased as the temperature increased, mostly due to the decrease in the amount of ferromagnetic *α′*-bcc martensite, but also due to the thermal effect that prevented the magnetic moments to follow the direction of the applied magnetic field.

The effects of applied magnetic field on magnetic transition are illustrated in Figure 3 by means of normalized magnetization curves versus the temperature for the samples *20_MA_700* and *40_MA_700*.

The magnetization first increased up to a maximum observed on all *M/M_max_* curves, due to the fact that the magnetic moments followed the direction of the magnetic field, thereafter decreased as the thermal effect became predominant. The temperature values corresponding to the *M/M_max_* maximum of the curves represent the moments where thermal movement completely disoriented the magnetic spins. These values were smaller for samples subjected to higher applied magnetic fields since the higher the applied magnetic field the easier the alignment of the magnetic moments. This confirms our assumption that the low values of applied magnetic field retarded this magnetic transition. The lowest temperatures of magnetic spin misalignment were 69 °C, for *20_MA_700*, and 52 °C, for *40_MA_700*, corresponding to an applied magnetic field of 1 kOe.

Considering the key role played by magnetism for the phase stability of Fe-Mn based alloys [32], the morphological particularities of *α′*-bcc martensite were further studied.

### 3.2. Structural Analysis

Some characteristic structural aspects are revealed by means of the SEM micrographs shown in Figure 4.

Martensite plates are present in the initial structure of each of the six specimens under study, which confirms the predicted martensitic structure, assumed in the discussion of Figure 1. In addition, most of the plates are short and have a marked surface relief that is typical for *α′*-bcc martensite [33]. These characteristics are more obvious in Figure 4b, which illustrates the microstructure of specimen *0_MA_1100*, and Figure 4f, corresponding to specimen *40_MA_1100*. These aspects prove that MA effects in FeMnSiCrNi are less intense than those observed in TiNbCuNiAl [34] or in TiFeCoNbAl [35].

However, some characteristic morphological aspects, such as the thin straight plates, reveal the presence of *ε*–hcp martensite. It is noticeable that, even if it is absent in the initial solution treated condition, *ε*–hcp martensite can be stress-induced by mechanical bending caused by the DMA push rod. Unlike the formation of stable statically stress-induced martensite, which is the pre-requisite of SME occurrence in any SMA, including FeMnSiCrNi [36], the dynamically stress-induced martensite forms as an effect of any shape change, induced by pushrod, even at very small amplitudes. The formation of stress-induced martensite has been associated with the presence of storage modulus plateaus on the DMA diagrams recorded during isothermal strain sweeps [20]. The presence of free recovery SME and its improvement after ten training cycles is shown in the Appendix A.

For the better observation of the particularities of *α′*-bcc martensite plates and intercrystalline interfaces, HR-SEM micrographs were recorded on specimens *0_MA_1100* and *40_MA_1100*, as illustrated in the typical micrographs shown in Figure 5.

At specimen *0_MA_1100,* it has been previously argued that the presence of 20 nm spaced internal sub-bands can be the cause of hindering the formation and growth of α′ stress-induced martensite [20]. Five parallel sub-bands are visible in Figure 5a. These sub-bands, which have been observed in the substructure of thermally induced α′ martensite [21], are much thinner than *ε*–hcp martensite plates that displayed a typical width of 100 nm [19]. Since three sub-bands are noticeable in the width of a single α′ martensite plate in Figure 5b, considering a large number of such plates, their cumulated presence might contribute to hindering the martensite reversion to austenite during heating, causing blocking of martensite/austenite interface movement. Consequently, when the interface finally crossed these sub-bands, caused an instability associated with a temporary storage modulus increase. In the case of specimen *0_MA_1100*, *ΔE′* ≈ 4.2 GPa.

On the other hand, at specimen *40_MA_1100*, the storage modulus increase, Δ*E′*, did not exceed 2.7 GPa. Figure 5c illustrates an intergranular crack formed along grain boundaries [19]. Its width, emphasized in Figure 5d, was 180 nm and could be the cause of the small storage modulus increase, as compared to other specimens.

To ascertain the phase structure predicted by SEM in Figure 4, XRD patterns were recorded, as previously discussed [19,20,21]. The typical structural aspects are exemplified in Figure 6, for the specimens that were solution-treated at 700 °C.

It was noticed that increasing the solution treatment temperature to 1100 °C caused a decrease in the ε–hcp martensite fraction, as reported in the literature, in the case of an FeMnSiCrNi SMA with a chemical composition of 0.05 mass% C, 12.60 mass% Mn, 6.00 mass% Si, 9.27 mass% Cr, 4.74 mass% Ni, 0.27 mass% Al and 0.13 mass% Mo [37].

As previously pointed out, the relative amounts of the three constitutive phases were determined by Rietveld analysis using the non-overlapping diffraction maxima, α′ (110), α′ (200), ε (101), γ (200) and γ (222) [20].The massive presence of martensite is noticeable. *0_MA_700* specimen contains both *α′*-bcc (47%) and ε–hcp (16%) martensites under various crystallographic plate variants (the rest of 37% corresponds to *γ*–*fcc* austenite). With increasing MA fraction, the amount of ε–hcp slightly increased to 23%, *α′*-bcc decreased to 22%, while *γ–fcc* augmented to 50%. A large amount of austenite, in the 20_MA_700 specimen, could be an explanation for its internal friction peak, which has the largest value, tan δ = 0.045. Finally, specimen 40_MA_700 comprised an amount of 82% *α′*-bcc, 14% ε–hcp and 12% austenite. The small amount of ε–hcp martensite, in this specimen, could be an explanation for the absence of the second tanδ maximum in Figure 1e. These data prove the presence of *α′*-bcc martensite in the initial state of all the specimens under study.

When summarizing the correspondence between storage modulus increase and the amounts of *α′*-bcc martensite and *γ*–*fcc* austenite, Figure 7 is obtained.

It can be observed that the highest storage modulus increases occurred in the specimens that had the largest amounts of austenite (50–55%) and the lowest amounts of *α′*-bcc martensite (22–37%). Since the specimens with the largest amounts of austenite experienced the largest storage modulus increases, it can be assumed that magnetic transition occurred to a larger extent in these specimens, causing the most important destabilizing of magnetic moments ordering. Thus, we assume that storage modulus increase could be related to the pre-existence of both martensite reversion and ferromagnetic *α′*-bcc martensite.

## 4. Conclusions

Summarizing the above discussion, the following conclusions can be drawn:Two internal friction maxima (*tan δ_max_* = 0.045) and a sudden storage modulus increase (*ΔE′_max_* = 12.2 GPa) were noticed on the DMA thermograms recorded during the heating, between RT and 400 °C, of solution-treated PM-MA FeMnSiCrNi SMAs.The thermal range of *ΔE′* had the tendency to increase with solution treatment temperature, its occurrence being associated with the presence of internal sub-bands within *α′*-bcc martensite plates.A magnetic transition, associated with magnetic spin temporary alignment, was observed by thermomagnetic measurements. The temperature of maximum magnetization corresponds to the point where thermal movement became prevalent over magnetic spin alignment. Its value had the tendency to decrease, from 405 °C to 52 °C, with increasing both the intensity of applied magnetic field, from 20 Oe to 1 kOe, and the fraction of mechanically alloyed powders, from 20 vol % to 40 vol %.At room temperature, the microstructure of the solution-treated specimens comprised *α′*-bcc martensite (22–82%), a limited fraction of ε–hcp martensite (5–23%) and retained austenite.Martensite reversion to austenite was proved by emphasizing free-recovery shape memory effect, the rate of which augmented after ten training cycles.A correlation was proposed between storage modulus increase, the pre-existence of γ–*fcc* austenite, martensite reversion and the destabilization caused by temporary magnetic spin alignment.

## Figures and Tables

**Figure 1 nanomaterials-12-02342-f001:**
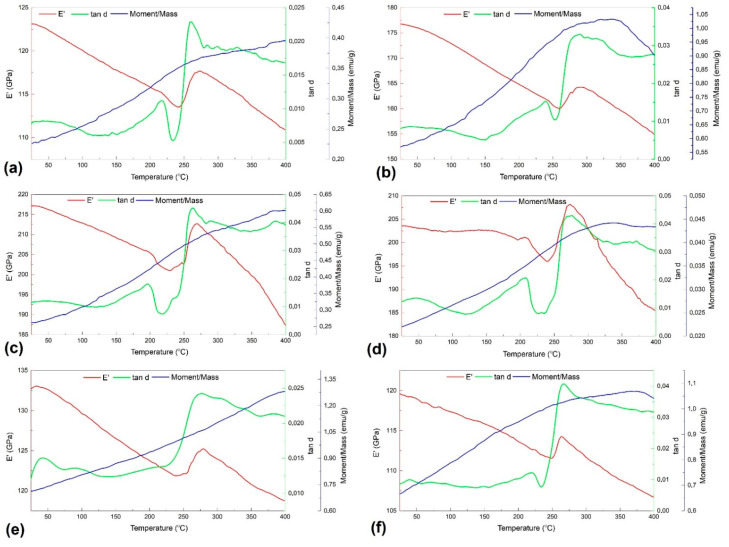
Thermograms recorded during heating, illustrating the variations with temperature of *E′*, *tan δ* and *M* for an applied magnetic field of 20 Oe, in the case six of specimens: (**a**) *0_MA_700*; (**b**) *0_MA_1100*; (**c**) *20_MA_700*; (**d**) *20_MA_1100*; (**e**) *40_MA_700*; (**f**) *40_MA_1100*.

**Figure 2 nanomaterials-12-02342-f002:**
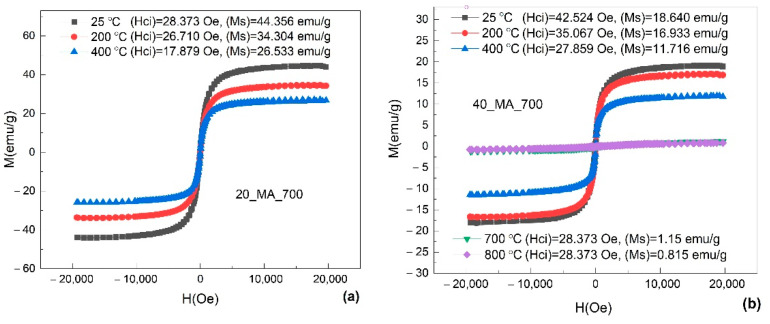
Magnetic hysteresis loops for two samples, at different temperatures: (**a**) *20_MA_700* at three temperatures: 25 °C, 200 °C and 400 °C and (**b**) *40_MA_700* at 25 °C, 200 °C, 400 °C, 700 °C and 800 °C.

**Figure 3 nanomaterials-12-02342-f003:**
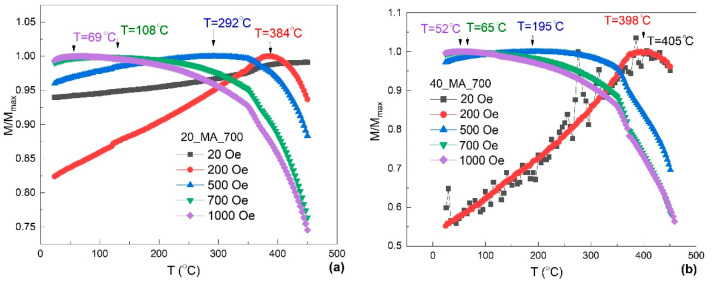
Normalized magnetization curves versus temperature for the samples (**a**) *20-MA-700* and (**b**) *40_MA_700*, illustrating the temperatures of the magnetization maxima.

**Figure 4 nanomaterials-12-02342-f004:**
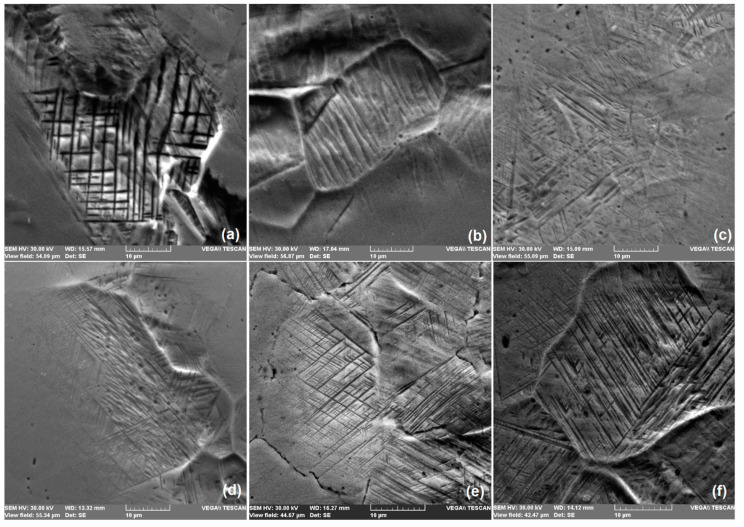
SEM micrographs in the solution-treated state: (**a**) *0_MA_700*; (**b**) *0_MA_1100*; (**c**) *20_MA_700*; (**d**) *20_MA_1100*; (**e**) *40_MA_700*; (**f**) *40_MA_1100*.

**Figure 5 nanomaterials-12-02342-f005:**
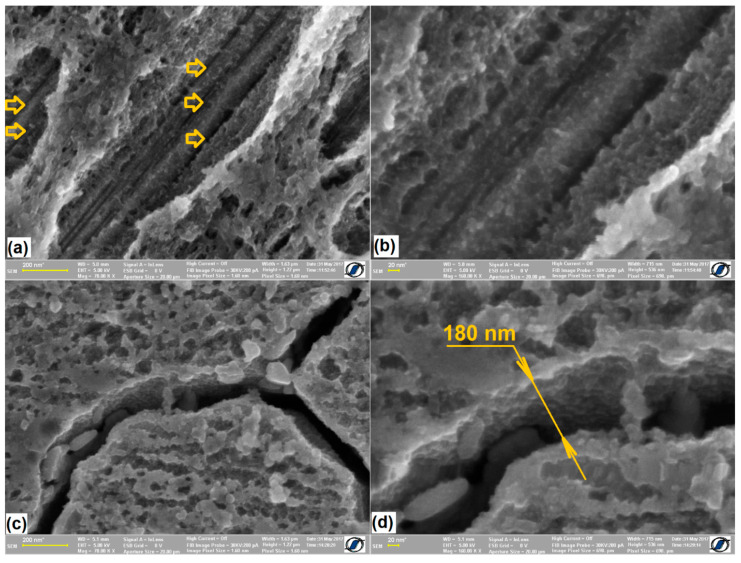
HR-SEM micrographs of specimens: (**a**) *0_MA_1100*, with a *α′*-bcc martensite plate comprising internal sub-bands marked by yellow arrows; (**b**) detail of three sub-bands disposed within the width of a single *α′*-bcc martensite plate, at *0_MA_1100*; (**c**) intergranular crack at *40_MA_1100*; (**d**) detail of the intergranular crack width, at *40_MA_1100*.

**Figure 6 nanomaterials-12-02342-f006:**
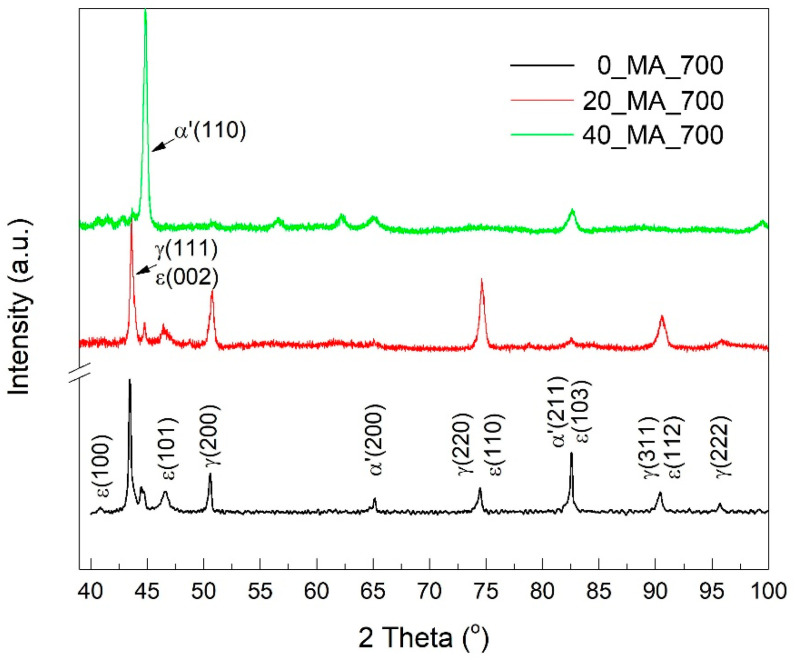
XRD patterns of the specimens solution-treated at 700 °C.

**Figure 7 nanomaterials-12-02342-f007:**
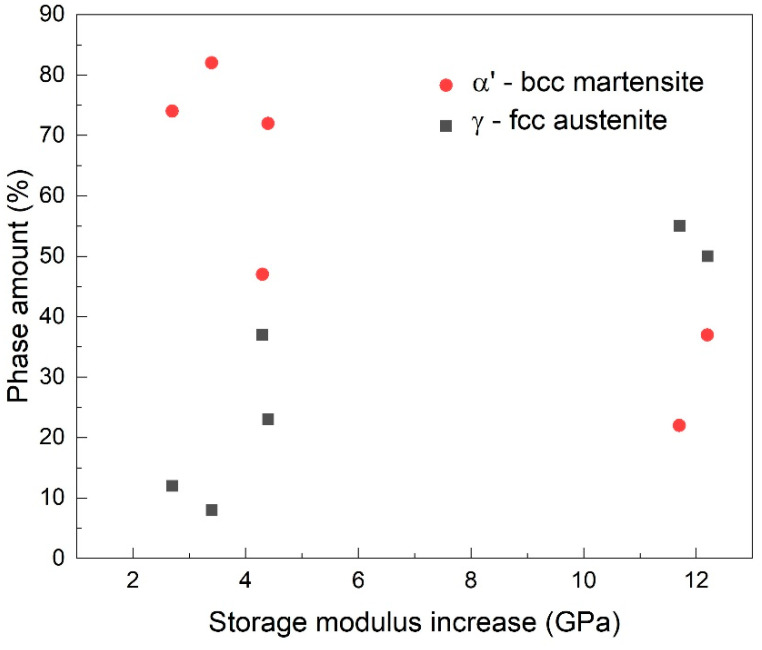
The relationship between storage modulus increase and the amounts of *α′*-bcc martensite and γ–*fcc* austenite.

## Data Availability

Not applicable.

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
