# Peer review of "On the Possible Cause of Sudden Storage Modulus Increase during the Heating of PM FeMnSiCrNi SMAs"

_nanomaterials, 2022, doi:10.3390/nano12142342_

Round 1
Reviewer 1 Report
In this work a series of experimental characterization are presented to describe and explain the behavior of IF parameter and Storage Modulus increase in temperature and by application of Magnetic field.
I think that the results must be explained better particularly concerning the DMA results:
1) which is the strain value applied during the measurements?
2) please the authors more explain if and how it is possible to have strain induced martensite at that strain
3) line 157: how this aspect considers or explains the more mechanical dissipation and intrinsic higher value of IF everytime reported in literature for the martensite?
4) please describe better which peak of X-ray diffraction are considered to evaluate the percentage of A or M content
Reviewer 2 Report
In this work, the antiferromagnetic-paramagnetic transition and martensitic transformation were examined through T-MAG and DMA. The reason for sudden storage modulus increases during heating was well analyzed. A major revision is suggested before it can be considered for publication in nanomaterials.
1. Page 2 line 72. The decreasing tendency of ΔE’ intensity with the increase of solution treatment temperature in Ref. 18 is contrary to the description in abstract. It needs clarify.
2. Page 3 line 131, “ electron beam resolution of 1.1÷2.5 nm ”, the label is wrong.
3. Why were different heating rates selected for T-MAG and DMA experiments? Can't it be consistent?
4. For magnetic hysteresis loops examination, why only were the samples 20_MA_700 and 40_MA_700 chosen? Reasons need to be explained.
5. “HR-SEM” The first occurrence of the word should be written in full.
6. Page 8. The calculation method for amount of each phase should be specified.
7. There are many problems in the figures, which need to be further checked and improved.
Figure 1 (e) “E’ (GPa)” is covered.
The legends in Figure 2. a b and Figure 3. a b are different in word size and border.
The name of sample should also be added in Figure 3. b like that in Figure 3. A.
The positions of axis title in Figure 3 are different.
The names of samples in Figure 6 should be the same as above.
8. For Figure 7, a bar chart is more recommended to express the amounts of martensite and austenite.
9. In this paper, samples containing different amount of MA powders were fabricated. It would be better to compare and discuss the difference of the resultant microstructure and the corresponding effect on the antiferromagnetic-paramagnetic transition and martensitic transformation. Here are some related articles which the author could refer to.
Materials Science and Engineering A, 700 (2017) 10–18.
Materials Science and Engineering A, 580, (2013) 397–405.
Reviewer 3 Report
The paper titled “On the possible cause of sudden storage modulus increase during the heating of PM FeMnSiCrNi SMAs” is aimed to clarify the effect of antiferromagnetic-paramagnetic transition on reverse martensitic transformations of ε - hcp and α’- bcc martensites. The manuscript is well arranged and reports new experimental data on SME of the FeMnSiCrNi alloy. I recommend this paper for publication after minor revision. The comments are listed below.
1. More information regarding the high-energy ball mill and the turbula blender should be given.
2. Is there any iron coming from stainless steel milling balls during the MA procedure?
3. Line 269: “in the initial state of all al the specimens” – What is “al”?
4. The description of the Supplementary Materials (lines 305-306) should be corrected. There are two videos, not Figure S1, Table S1, Video S1.
Round 2
Reviewer 1 Report
The authors addressed correctly all my requests. Now the paper is suitable for publication
Reviewer 2 Report
It is suggested that the author make conclusions brief and polish the language.